# The Concept of Integration between State and Provincial Sea Boundaries in Indonesia

Eka Djunarsjah [1,2,*], Andika Permadi Putra [1,2], Difa Kusumadewi [2], Kevin Yudistira [1] and Miga Magenika Julian [1,2]

1    Geodesy and Geomatic Engineering, Bandung Institute of Technology, Bandung 40132, West Java, Indonesia; andikapermadi03@gmail.com (A.P.P.); kevinyudis98@gmail.com (K.Y.); miga@gd.itb.ac.id (M.M.J.)
2    Hydrography Research Group, Bandung Institute of Technology, Bandung 40132, West Java, Indonesia; kusumadewi.difa@gmail.com
*    Correspondence: lautaneka@gmail.com

**Abstract:** The clarity of marine spatial status requires a coherency between the state (territorial) sea boundary and the provincial sea boundary because both have the same sea width of 12 nautical miles. However, the two sea boundaries use different references; the state sea boundary refers to the low-water line, whereas the provincial sea boundary refers to the high-water line, so that the outer limits of the two sea boundaries differ. From the applicable provisions, the provincial sea management area may not exceed the territorial sea area. The method used to detect the overlap between the state sea and the provincial sea is the overlay method. By taking the study area of the waters of the province of East Nusa Tenggara, a difference is produced between the position of the outermost boundary of the territorial sea of the state of Indonesia and the sea area of East Nusa Tenggara province. The results of this study show differences in the boundaries of the state and provincial seas. To prevent potential issues, in the process of implementing Marine Spatial Planning (MSP) and/or Integrated Coastal Zone Management (ICZM), there must be a clarity and a uniform reference between the state and provincial sea boundaries; otherwise, it will create an overlap of authorities, between the state (central) and the provincial level. This incoherency shows that this practice has not fulfilled one of the requirements in implementing ICZM, which requires the integration of all aspects, including political boundaries.

**Keywords:** boundary; state; province; MSP; ICZM





## 1. Introduction

The coastal zone is an area where the topography of the land borders the sea. The coastal zone is affected by natural phenomena from both the land and the sea. The coastal area covers parts of dry land and land that is submerged in water, which is influenced by ocean dynamics, such as tides, oceanic waves, and saltwater infiltration. Towards the sea, the coastal area includes parts of the sea that are still affected by natural processes that occur on land, such as sedimentation, the hydrologic cycle, and human activities in the form of deforestation and pollution [1]. In Law Number 1/2014, concerning the Zoning Plan for coastal areas and small islands, the coastal area is defined as the land boundary of the administrative area at the temporary sub-district level to the sea, at a distance of 12 nautical miles from the coastline [2].

Activities in the coastal area are managed by cross-sectoral institutions, each of which is interested in carrying out their activities. The task of managing coastal areas in Indonesia is carried out simultaneously by multiple institutions with overlapping job descriptions. If seen from a practical aspect, there are two conditions that can pose a problem regarding the management of coastal areas; one is that it has multiple and redundant organizations managing it, and one is that it has no management or supervision whatsoever. The most common issue that arises with this condition is that it is probable to have a condition

of overlapping jurisdiction in a single coastal area, which may lead to the imposition of different legislative rules for the area, depending on the governing institution. For example, a strategic part of the sea and its adjacent coastline may be a site of interest for organizations that might want to install an undersea energy pipeline or communication cables, while other organizations might want to create a clear, non-obstructive zone within the waters to create a strategic sailing route for sea transportation vessels. Some of these uses may be counterproductive, if combined within a common spatial scope, so it is important to create a certainty of rights, responsibilities, and regulations that bind the stakeholders in the maritime sector [3].

In its implementation, there are sectoral contestations of control of coastal natural resources and environmental services. This is due to overlapping sectoral legislation, as the task of managing coastal areas in Indonesia is carried out simultaneously by multiple institutions with overlapping job descriptions. An example of this issue is that of the Ministry of Tourism and Creative Economy, Ministry of Energy and Mineral Resources, and the Ministry of Environment and Forestry, which all have different interests in utilizing resources in coastal areas. The Ministry of Tourism and Creative Economy uses the available land area in and nearby coastal areas to build tourism-supporting infrastructure. The Ministry of Energy and Mineral Resources will seek to develop renewable energy in coastal areas, while, the Ministry of Environment and Forestry seeks to maintain the quality of the environment and ecosystems in coastal areas.

A lack of understanding of the stakeholders in coastal areas could inhibit the development and growth of the area and the local community. Without adequate knowledge, the site executors of the stakeholders tend to focus on doing only their organizational tasks, without opening up to the possibility of cooperation and doing further development of the area with other stakeholders. This is detrimental to the area and the local community, and it creates a "sectoral ego" that leads to a rift between each individual stakeholder. In each sector's condition, in implementing their respective procedures, there is the potential for conflict in managing natural resources and services in coastal areas. The lack of understanding of stakeholders in the functions of the coastal zone will cause inappropriate usage of the area. An example of the issue is to place two areas of different zoning purposes next to each other, without any consideration of what implications each zonal area will impose on each other. A more accurate representation of this scheme is the existence of a tourism development zone that is situated next to an aquaculture fisheries zone. Currently, the coastal areas in Indonesia are regulated by many laws and multiple institutions. This overlap and redundancy of the legal aspect can create a state in which a certain stakeholder may claim to have legitimacy or entitlement to exercise their own rights, even when it interferes or overlaps with those of other organizations. Therefore, it is necessary to map spatial conflicts in coastal areas. Mapping the potential for conflict is fundamental for analysis in drafting a zoning plan for coastal areas and small islands. Knowing of the potential conflicts regarding spatial use activities, solutions and future policy directions can be anticipated [3].

In implementing the division of regional sea authority and state seas, the United States, with its federal system, provides boundaries for regional seas in areas still included in the United States territorial sea. In addition to referring to international law, each federal state with a coast can use its influence in the United States territorial sea [4]. Indonesia, which implements regional autonomy, needs to emphasize clear boundaries between territorial seas and regional seas to reduce problems arising from biased management of marine areas.

A previous study, regarding the implementation of Integrated Coastal Zone Management (ICZM), was carried out in the Mediterranean Sea, involving multiple sovereign states. Several issues have arisen that inhibit the process of implementing the ICZM concept, such as the lack of financial resources, the lack of a proper evaluation and supervision system, the lack of knowledge and expertise on coastal systems, the lack of high-quality human resources, as well as the lack of public participation and administration of information-based strategy [5].

A similar study has also been carried out in the Adriatic Sea, where an evaluation process on managing coastal areas was done by doing Marine Spatial Planning (MSP) [6]. In the context of delineating sea boundaries of sovereign state and intra-state administrative divisions, the United States of America has implemented a uniform baseline, which refers to the line formed by the lowest water level [7].

The comparison of Indonesia and the United States of America, related to the reference for determining boundaries, is described in Table 1.

**Table 1.** Reference Comparison of Indonesia and United States of America.

| Number | Object | Indonesia | US |
|---|---|---|---|
| 1 | Chart Datum | Mean Low Water Spring | Mean Lower Low Water and Mean Low Water |
| 2 | State Sea Boundary Reference | Low Water Line | Low Water Line |
| 3 | Province Sea/Federal Government Boundary Reference | High Water Line | Low Water Line |

Source: Compiled by the author.

In conjunction with the legal aspect, the laws that govern provincial and national authorities are within the same hierarchical branch. Law Number 6/1996, concerning Indonesian Waters, defines the territorial waters of Indonesia as a whole, while Law Number 23/2014, concerning regional government, defines the sea boundaries for the first level administrative divisions (provinces) of Indonesia. To determine the national maritime (state sea) boundary, Indonesia uses the baseline as its basis, where the baseline is generated by the low-water level line. Meanwhile, in determining sea boundaries between provinces within Indonesia, the high-water level line is used as the basis. These two water levels are coastline, as in the perspective of Law number 4/2011, concerning geospatial information, and the coastline at the Mean Sea Level (MSL).

Table 2, below, describes the various stakeholders in the Indonesian maritime sector, with its corresponding legislation, regarding the technical aspects of determining sea boundaries.

**Table 2.** Indonesian Stakeholders for Determining Sea Boundaries.

| Number | Ministry/Institution | Coastline | Legal Basis |
|---|---|---|---|
| 1 | Centre of Hydrography and Oceanography of the Indonesian Navy | The coastline that is the basis of determining state sea boundaries is the surface of seawater at its lowest tide. | Government Regulation number 38/2002, later amended by Government Regulation number 37/2008 |
| 2 | Ministry of Internal Affairs | The coastline that is the basis of determining province sea boundaries is the surface of seawater at its highest tide | Minister of Internal Affairs Decree number 141/2017 |
| 3 | Government of East Nusa Tenggara Province | The coastline that is the basis of determining province sea boundaries is the surface of seawater at its highest tide | East Nusa Tenggara Provincial Government Regulation number 4/2017 |

Source: Compiled by the author.

This study aims to understand what part integration plays in determining province and state sea boundaries in Indonesia. Thus, the focuses of the research of this study, seen below, are:

1.  How to determine the provincial sea boundaries that are integrated with state sea boundaries.
2.  The impact of incoherence between province and state sea boundaries.

## 2. Materials and Methods

This study location is in the province of East Nusa Tenggara with an integration study between provincial and state sea boundaries. The study area is presented in Figure 1.

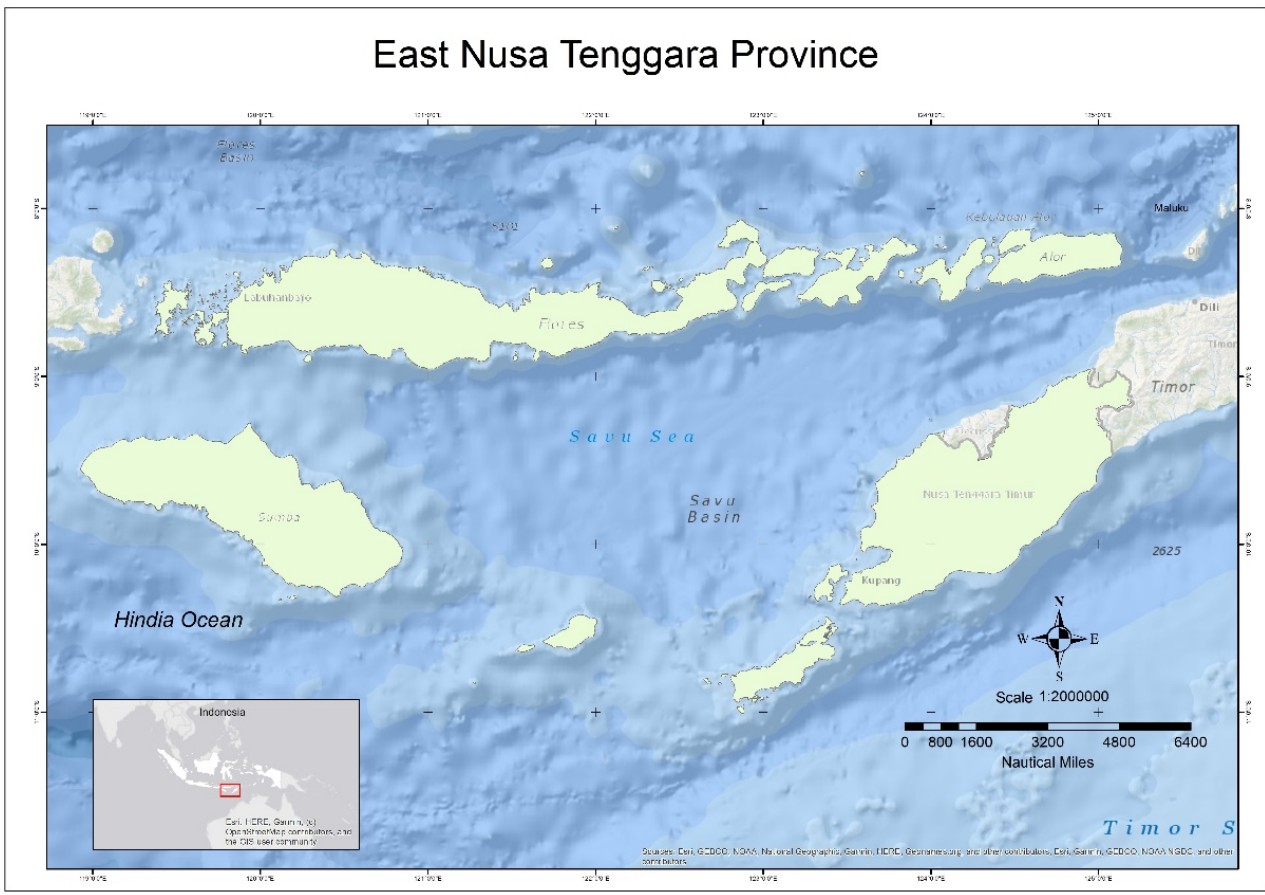

**Figure 1.** Study Area.

East Nusa Tenggara province is an archipelagic province that comprises the main islands of "Flobamorata", which is an acronym of the five main islands of the province, made up of the following: Flores, Sumba, Timor, Alor, and Lembata. East Nusa Tenggara province has a land area of 47,931.54 km$^2$, with the island of Timor as its largest island (14,732.35 km$^2$). As of 2020, East Nusa Tenggara province is further divided into administrative divisions, namely 21 regencies and one city. The regency with the largest land area is East Sumba Regency, followed by Kupang Regency, with areas of 7005.00 km$^2$ (14.61%) and 5525.83 km$^2$ (11.53%), respectively. The smallest area is Kupang City, 180.27 km$^2$ (0.38%). As an archipelagic province, there are several ways to reach Kupang, the capital city of the province. Regencies and cities located on the island of Timor (Kupang Regency, South Central Timor, North Central Timor, Belu, Malacca, and Kupang City) have land-based transportation networks that are connected to the capital. Meanwhile, other regencies rely on sea or air transportation [8].

The spatial data and information acquired on areas of interest for this study are derived from the Indonesian Baseline Chart, the Zoning Plan for Coastal Areas and small islands map of East Nusa Tenggara province, as well as a map of Economic Exclusive Zone (EEZ) number 8. The Zoning Plan of Coastal Areas and small islands map of the East Nusa Tenggara Province is formalized by Provincial Government Regulation Number 4/2017, concerning Marine Spatial Planning. This given area is chosen as the study area of interest because the province is an archipelagic province that, apart from still having intra-province sea boundaries to be resolved, borders two sovereign states (East Timor and Australia), which to an extent escalates the issue of sea boundary uncertainty in the province to an international level.

From a spatial perspective, the height of a position needs to be determined or defined so that it can be a reference to other positions. Respect to this position can be to the center of the earth, mean sea level, satellite orbit, or other more straightforward concerns. The height selected as a reference is referred to as the vertical datum [9]. At this time, there are 100–200 altitude references. In determining the boundary, the respect for the withdrawal is based on a baseline determined from the meeting between coastal topography and low-water level. The baseline itself is the locus of the basepoint, as shown in Figure 2.

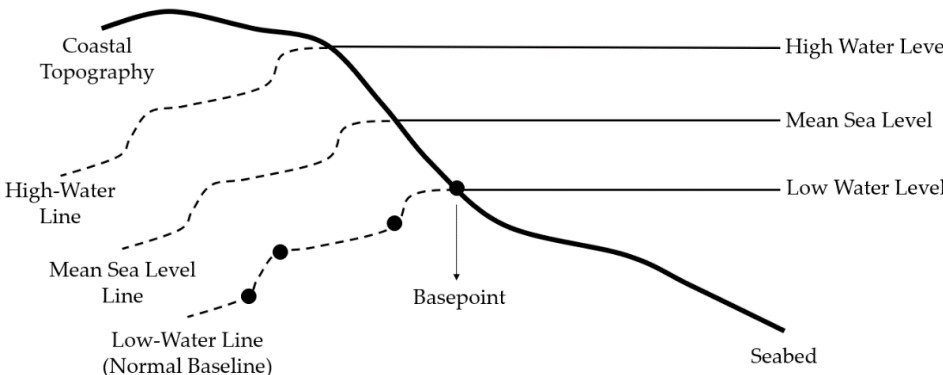

**Figure 2.** Baseline and Basepoint Illustration.

In Indonesia, the definition of a coastline is provided by Law Number 4/2011. The rule states that there are three types of coastline; namely, the high-water level line, the mean sea level line, and the low-water level line. The high-water level line is defined as the meeting between the coastal topography and the surface of the sea at its highest tide level. The mean sea level line is defined as the meeting between the coastal topography and the surface of the sea at its average surface height. Meanwhile, the low-water level is defined as the meeting between the coastal topography and the surface of the sea at its lowest tide level [10]. In conjunction with the definition of state sea boundaries (territorial sea) that refer to the UNCLOS III (1973–1982), it has been further formalized in Law Number 6/1996. In that Law, Indonesia uses the low-water level line as its baseline, while Law Number 23/2014 uses the high-water level line as the basis for determining sea boundaries between intra-state administrative divisions (provinces) of Indonesia [11].

Technical aspects are used to support Marine Spatial Planning (MSP), critical parameters used as references in determining boundaries, height references, or to detect physical elements of seawater that will affect zoning determination of marine areas. MSP is a guideline for managing and improving the marine environment's quality, which can be applied in research and practical fields. The MSP contribution scope includes marine zoning, defining boundaries, planning in a dynamic environment, stakeholder involvement, information needs, integrated marine and land use management, management of overlaps and conflicts of use, and cross-border institutional structures [12].

In another definition, MSP is the process of analyzing and allocating the spatial and temporal distribution of human activities in marine areas to achieve ecological, economic, and social objectives that have been specified through a political process. MSP is a practical

way to enhance the utilization of spaces on the sea surface, improve its productivity, create a balance between economic growth and environmental sustainability, and deliver social and economic outcomes for the local and regional community [13].

The Marine Spatial Plan in Indonesia, based on Government Regulation Number 32/2019, serves as a guideline for the formulation of the long-term national development plan in the Marine sector, the formulation of the mid-term national development plan in the Marine sector, the realization of development integrity and harmony, as well as cross-sectoral and interregional interests in utilizing and controlling marine spatial use nationally, the determination of location and spatial function for strategic or national priority activities, formulation of a zoning plan for sea area, and formulation of a zoning plan for coastal areas and small islands [14].

Article 6 of Law Number 26/2007, concerning Spatial Management among other things, states that the practice of spatial management within Indonesia includes exerting sovereignty over an integral territory that comprises land, sea, and air jurisdictions, as well as the resources that are contained within the earth under Indonesian land territory. The management of air and sea space, however, is further regulated by a separate law. As stated in Article 43, Section (2) of Law Number 32/2014, concerning Marine Affairs, the practice of marine spatial planning includes devising zoning plans for both coastal areas and small islands, as well as the sea surface itself. Marine Spatial Planning covers the waters area and jurisdiction area. Marine Spatial Planning for water areas includes arrangements related to policies and strategies for marine spatial management, marine spatial structure plan, marine spatial pattern plan, direction for marine spatial utilization, and direction for controlling marine spatial utilization in internal waters, archipelagic waters, and territorial sea. However, Marine Spatial Planning for the jurisdiction area includes policies and strategies for marine spatial management, marine spatial structure plan, marine spatial pattern plan, direction for marine spatial utilization, and direction for controlling marine spatial utilization in the Exclusive Economic Zone and continental shelf [15].

In a more detailed discussion, coastal area management is included in the concept of ICZM [16]. This concept has developed since 1970 and has become part of international conventions and treaties as a framework and guidance for area management at a local scale [16]. ICZM is part of the increased success in managing environmental resources that depend on a transition from a sectoral management approach to a structural policy that is more sensitive to "natural systems" [16]. It is further explained that ICZM is a dynamic process in developing and implementing a coordination strategy in allocating resources to achieve conservation and sustainable use of coastal areas [16]. To achieve such coordination, ICZM covers core management principles, including internal management:

1. With and across sectors so that sectoral activities sustainably take place.
2. With and between units and levels of government, including between states and the international community.
3. Cross time scales.
4. Between science and management.
5. Through the natural system and land–sea traffic [16].

One of the concepts regarding ICZM that was formulated during the 1992 Earth Summit in Rio de Janeiro, Brazil is that ICZM is defined as a coastal management process for the management of the coast using an integrated approach, regarding all aspects of the coastal zone, including geographical and political boundaries, in an attempt to achieve sustainability. This study shows that in the process of making the Zoning Plan of Coastal Areas and small islands map of East Nusa Tenggara province, there was a failure to consider political boundaries. It is indicated by the discrepancy of the foundation used by the Provincial Government in East Nusa Tenggara and the Indonesian Central Government in delineating both of their own sea boundaries. This research discusses the implications that result from not considering political boundaries in the process of implementing the ICZM [7]. The following diagram in Figure 3 explains the role of ICZM in this study:

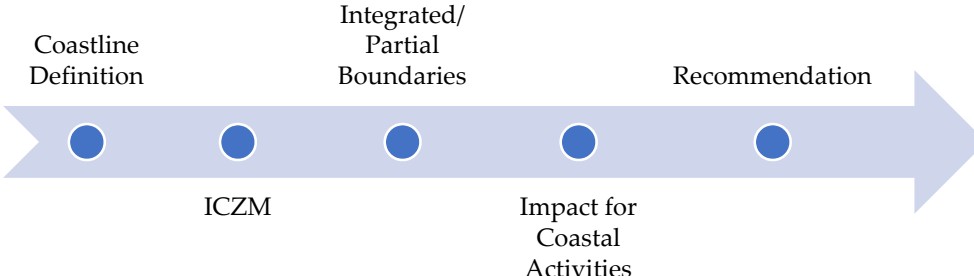

**Figure 3.** Research Flowchart.

The research regarding the integration of provincial and state sea boundaries encompasses the discussion of a coastline that serves as the reference in delineating the boundary. The process of boundary delineation that applies in a certain state is dependent on the set of regulations that are in place. The definition of the coastline in the law is of dire urgency as the difference of usage will create a spatial discrepancy on-site.

Furthermore, the study is concluded by discussing ICZM to ensure the activities that are carried out in coastal areas are sustainable. Referring to the regulations that apply in Indonesia, the delineation of provincial and state sea boundaries are defined by different sets of rules. The technical aspect of the national sea boundary is provided by Government Regulation Number 38/2002 concerning the List of Geographical Coordinates of Archipelagic Basepoints of Indonesia, as well as the amending decree, Government Regulation Number 37/2008, which states that Indonesia uses the low-water level line as the basis in determining the state sea boundary. The specific basis (chart datum) mentioned in the regulation is the Mean Low Water Spring (MLWS). Meanwhile, the sea boundaries between provincial divisions in Indonesia are governed by Minister of Internal Affairs Decree Number 141/2017 concerning affirmation of province boundaries, which explains that the basis of the boundary delineation process is the high-water level line. The difference of bases that are legitimized by both laws will result in different ways of managing the boundaries on-site.

This study is aimed at finding the impacts of the discrepancies of boundaries in multiple levels of the governmental hierarchy in Indonesia, to the activities carried out in the coastal areas of Indonesia. This study also aims at giving solutions and recommendations to the currently ongoing issue.

The one map policy embodies efforts to integrate spatial data standards. The Geospatial Information Agency, as the institution in charge of spatial data in Indonesia, is following the mandate of Law Number 4/2011. With various data issued by multiple agencies, the Geospatial Information Agency is supervised by providing basic geospatial information and thematic geospatial information as references for generating geospatial data in other agencies.

One of the objectives of the one map policy is to support spatial planning. Geospatial data and information are critical to support the sustainable development of a state; they increase the provision of geospatial data and information. With the decentralization of action, local governments have an essential role in the implementation of spatial planning. The one map policy must be developed to empower local governments, both technically and in human resources, to properly carry out spatial planning [17]. In ideal conditions, Geospatial Information Agency support for realizing spatial planning is to provide maps that support renewable varying scale spatial planning, prepare guidelines for spatial mapping, and increase the capacity of local government human resources to carry out spatial planning.

This study uses a quantitative research methodology by conducting comparative analysis and correlation analysis. Correlation analysis is carried out on each planning domain's spatial data by calculating the area of empty and overlapping spaces that occur in coastal areas in one province. This data can be analyzed related to potential losses due

to overlapping or bordering areas but also the use of different space allocations. From the data processing results, a correlation analysis was carried out with legal and institutional aspects to further investigate the source of the problem of inconsistency in spatial planning in coastal areas. The research output is to select technical parameters that can be integrated to support optimal coastal area management.

## 3. Results

The problem of coastal areas can be viewed from the integration of legal aspects, apart from technical aspects, which are the main factors in regional planning. Our analysis shows several discrepancies in legislation that can cause conflicts during field implementation. Table 3 is a list of institutions that are stakeholders in the Indonesian maritime sector.

**Table 3.** Listing of the Stakeholders in the Indonesian Maritime Sector.

| Number | Ministry/Institution | Authority | Legal Basis |
|---|---|---|---|
| 1 | Ministry of Defence | Synchronise strategy and planning with the inventory of potential forces of the national defence. | Republic of Indonesian Government Regulations number 58/2015 Chapter 3 |
| 2 | Indonesian National Armed Forces | Uphold the supremacy of law and ensure safety and stability within the national sovereignty. | Republic of Indonesian Government Regulations number 10/2010 Chapter 7 |
| 3 | Indonesian National Police Force | Uphold the law and order within the communities that reside along the coast and off-shore of Indonesian seas. | Chief of Indonesian National Police Force Regulation number 21/2010 |
| 4 | Ministry of Transportation | Advise, create, and implement policies regarding shipping and naval transport. | Republic of Indonesian President Regulations number 40/2015 Chapter 13 |
| 5 | Ministry of Law and Human Rights Affairs | Organise the legal affairs of the multiple stakeholders of the Indonesian maritime sector. | Republic of Indonesian Government Regulations number 24/2010 |
| 6 | Ministry of Finance | Organise and create a financial scheme for the multiple stakeholders of the Indonesian maritime sector. | Republic of Indonesian Government Regulations number 28/2015 |
| 7 | Ministry of The Internal Affairs | Coordinate, facilitate, supervise, and evaluate the development of maritime transportation, conservation of the environment, spatial planning and management, and standardise the provision of energy and tourism service. | Minister of The Interior Regulations number 41/2010 Chapter 345, Chapter 405, Chapter 444, Chapter 448, and Chapter 597 |
| 8 | Marine Security Agency | Execute maritime safety operations that are supported by the Early Warning System and cooperate with law enforcement agencies to penalise offenders within the Indonesian maritime sovereignity. | Republic of Indonesian Government Regulations number 178/2014 |
| 9 | Ministry of Marine Affairs and Fisheries | Organise the spatial zoning on Indonesian waters, manage the conservation of marine biodiversity, organise capture fisheries, organise aquaculture fisheries, empower and enhance the competitiveness and logistics support system of the maritime and fisheries industry, ensure the long-term sustainability of maritime and fisheries industry. | Republic of Indonesian Government Regulations number 63/2015 Chapter 3 |

**Table 3.** *Cont.*

| Number | Ministry/Institution | Authority | Legal Basis |
|---|---|---|---|
| 10 | Geospatial Information Agency | Provide and manage geospatial data and information regarding the maritime sector in Indonesia | Chief of Geospatial Information Agency number 3/2012 Chapter3 |
| 11 | Land Administration Agency of Indonesia | Organise affairs related to land administration. | Chief of Land Administration Agency of Indonesia Regulation number 3/2006 Chapter 3 |
| 12 | Centre of Hydrography and Oceanography of the Indonesian Navy | Organise and execute surveys and sea mapping, conduct research, and ensure the safety of sea navigation in Indonesia. | Republic of Indonesian Government Regulations number 164/1960 |
| 13 | Ministry of Energy and Mineral Resources | Organise affairs related to the exploration and exploitation of minerals and other earthly resources in the sea. | Minister of Energy and Mineral Resources Regulations number 18/2010 Chapter 3 |
| 14 | Ministry of Forestry and the Environment | Organise affairs regarding the sustainability of forestry and environment, manage and conserve the environment and the ecosystem within, enhance the quality of the environment, regulate the handling of environmental contamination and pollution, mitigate the risks of environmental contamination and pollution, mitigate the risks of climate change, and penalise offenders that threaten the environment, forestry, and the ecosystem within. | Republic of Indonesian Government Regulations number 16/2015 Chapter 3 |
| 15 | Ministry of Tourism and Creative Economy | Organise affairs related to tourist destinations and the development of the tourism industry, market the Indonesian tourism industry domestically and internationally, and develop institutions related to the tourism sector. | Republic of Indonesian Government Regulations number 19/2015 Chapter 3 |

Source: Compiled by the author.

The integration of provincial and state sea boundaries is essential in identifying clear boundaries for provincial seas that intersect with the territorial sea. The impact of not integrating provincial and state sea boundaries results in overlapping authority between the central and provincial governments. For example, in East Nusa Tenggara province, the spatial data of the Zoning Plan of Coastal Areas and Small Islands exceeds the territorial sea boundary line on the Exclusive Economic Zone (EEZ) map. The following is presented in map form in Figure 4.

Based on data processing results, the area of zoning outside the territorial sea is 204.14 km$^2$. Details are shown in Table 4.

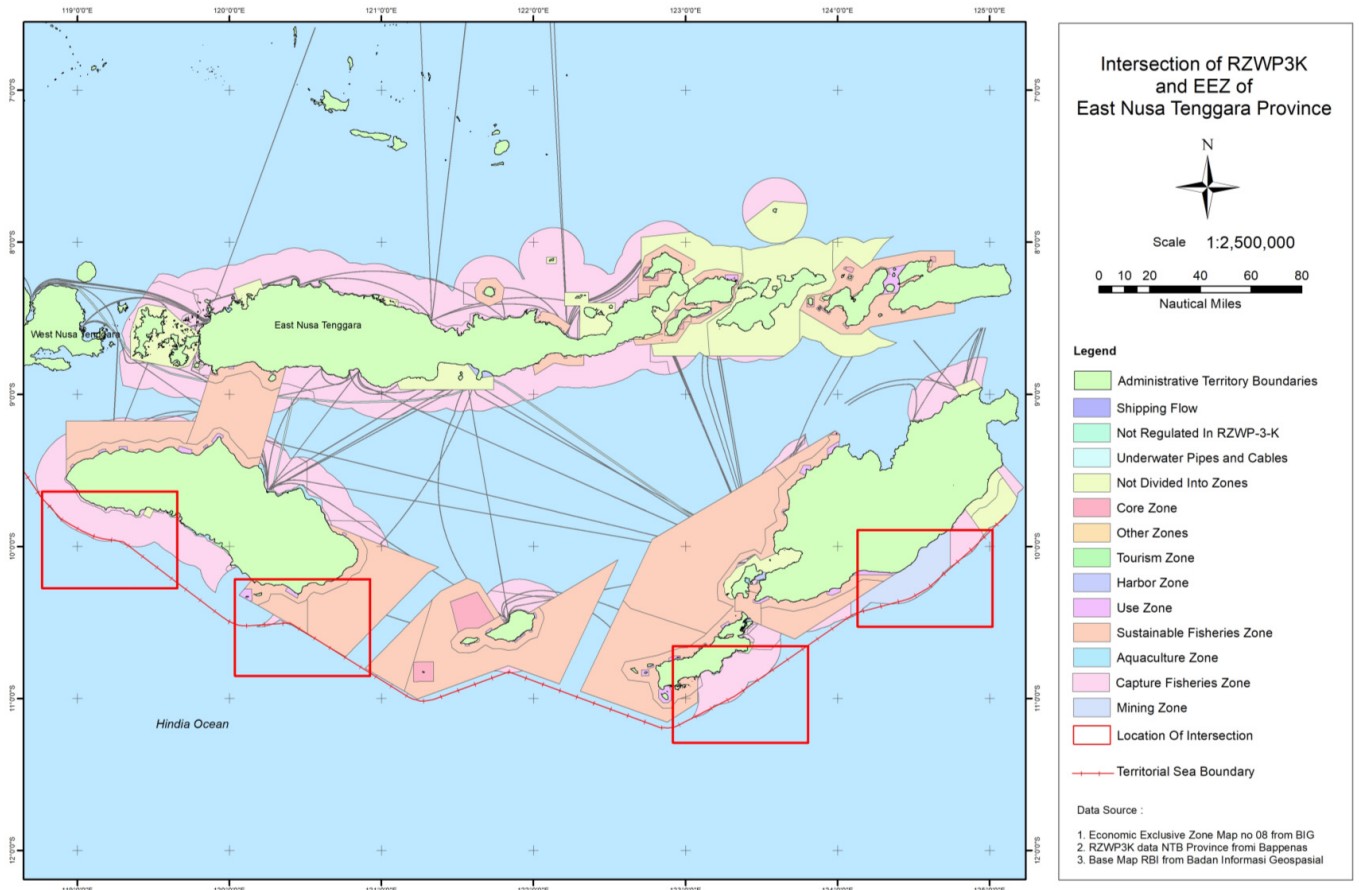

**Figure 4.** Intersection of the Provincial Sea Area and Territorial Sea.

There is the potential for abuse of local governments' authority in marine areas outside the territorial sea from the area obtained. Based on the provincial sea overlay with the territorial sea, 22 provinces are directly adjacent to the territorial sea so that 22 regions can provide overlapping authority between the Central Government and the Provincial Government.

The problem with the broader area than that of the territorial sea is caused by different data sources, including the boundary drawdown reference and the projection system. In ideal conditions, the territorial boundary cannot exceed the territorial sea because it uses the high-water line to reference boundary drawdown. In contrast, the territorial sea uses the low-water line. Horizontally, the 12 nautical mile claim for the territorial sea covers a more expansive ocean.

**Table 4.** The Provincial Sea Area outside the Territorial Sea.

| Number | Area | Zone | Location | Large Area (km$^2$) |
|---|---|---|---|---|
| 1 | Public Use Area | Mining Zone | South Central Timor Regency | 56.08 |
| 2 | Public Use Area | Capture Fisheries Zone | North Kodi District to Pinu Bahar District | 62.50 |
| 3 | Public Use Area | Capture Fisheries Zone | East of Boking District to Wewiku District | 15.62 |
| 4 | Public Use Area | Capture Fisheries Zone | South Amarasi District Waters | 2.63 |
| 5 | Public Use Area | Capture Fisheries Zone | South Salura Island Water | 9.91 |
| 6 | Public Use Area | Capture Fisheries Zone | Lobalain District to East Rote | 41.79 |
| 7 | Public Use Area | Capture Fisheries Zone | The Waters South of Sabu Island | 1.12 |
| 8 | Conservation Area | Sustainable Fisheries Zone | Southern Waters of Ngadu Ngala District | 7.95 |
| 9 | Conservation Area | Sustainable Fisheries Zone | The Eastern Waters of Wula Waijelu District | 1.32 |
| 10 | Conservation Area | Not Divided Into Zones | Southern Waters of Wewiku District | 5.22 |
| | | | Total Area | 204.14 |

Source: Compiled by the author.

## 4. Discussion

A correlation analysis was carried out on the previously discussed legal and institutional factors from studying the technical aspects explained. The technical aspects determination is very much influenced by policies at the formal and explicit coordination of the institutions related to coastal area management.

The technical aspects analysed relate to the one map policy. The technical elements that need to be reviewed are outlined below:

- Data Source Specifications.
- The Need for basic geospatial information and thematic geospatial information.
- Integration of the Spatial Pattern Classification System.
- Spatial mapping analysis.
- Map database and map of cartographical spatial map output.
- Integration of spatial planning and infrastructure network plans [18].

In this study, the technical aspects studied in the one map policy consisted of data source specifications, basic geospatial information and thematic geospatial information requirements, and zone classification system integration. To realize the one map policy in the context of coastal area management, sectoral agencies (ministries or institutions) and structural institutions (central and regional governments) need to formulate mutually agreed upon technical aspects for Marine Spatial Planning, such as necessary geospatial information that becomes a reference for determining boundaries, and a zoning determination mechanism in provincial border areas that will support the optimal utilization of one another.

Currently, the availability of basic geospatial information and thematic geospatial information still does not fully support spatial planning, due to several problems [17], as follows:

- Thematic geospatial information has an unclear existence and distribution in supporting spatial planning, which can affect the planning maps' speed nationally.
- Limited availability of thematic geospatial information to support spatial planning. This should provide a strong impetus to complete the lacking thematic geospatial information nationally.
- We have a limited guidance document for spatial planning mapping. The existence of a guideline document would accelerate the completion of spatial planning mapping.
- Lack of ministerial/institutional activities to accelerate the implementation of thematic geospatial information.

When the conditions still overlap due to various aspects, the authors recommend two solutions that can be instigated during the one map policy transition period, including:

- Prioritizing the boundaries of the territorial sea, which is the state's authority. After determining the state's territorial limits, the determination of provincial sea boundaries must be adjusted not to exceed the territorial sea boundaries. The technical aspects used to determine state boundaries must also be implemented to determine the provincial sea.
- In overlapping areas, joint management is carried out, with zoning adjustments agreed upon by the institutions that manage the provincial sea and the state sea to work the area.

Based on the analysis that has been carried out, technical problems still pose a challenge as one of the keys to realizing the integration of Marine Spatial Planning. Within the ICZM framework, the technical aspect is a function of the legal and institutional factors that become the legal aspect and the executor of authority in coastal areas. Analysing the correlation between technical, legal, and institutional aspects is very relevant for achieving an ICZM.

The technical aspects of the ICZM need to have measurable characteristics and can be statistically analysed. The determination of technical aspects can develop according to changes in the methodology used. An example of this is the judgment of the High-Water Mark (HWM) [19]. There is no agreed definition yet for the High-Water Mark, as this can depend on the diversity of beach types, the determination of objectives, and the limitations of observation techniques and data. The definition of HWM varies depending on the regulations and understandings adopted in a state. In other research, technical aspects will impact other political, economic, administrative, social, and environmental aspects. Thus, the integration of various elements is needed to reinforce what must be done in this field [20].

The technical aspects used in delineation can also contribute to developing a system that provides benefits for a sound registration system in the marine environment as a basis for legal certainty, with multiple services in the blue economic sector and Marine Spatial Planning [21]. For the case in Indonesia, a marine cadastral system's development is critical to support marine-oriented development.

The ICZM concept demands that all components and stakeholders that are involved in activities around the coastline are tightly integrated. The difference of authorities and rights between multiple organisations has the potential to overlap, making them redundant. The sea boundary is an integral part of the ICZM concept that represents the political boundaries that are set by political entities. With the implementation of decentralisation from the Central Government in Indonesia, the governing of rights and regulations is dependent on the provincial governments of Indonesia. In such conditions, coordination is necessary to make the ICZM come to fruition.

The discrepancy between the basis of boundary delineation at the province and state (central) levels shows that there has been an ongoing lack of coordination between the stakeholders, the policymakers, and the grassroots-level executors on-site. Referring to the data used, two differences can be considered in determining the sea boundaries of the provincial and state sea boundaries in Indonesia. The first difference is seen from the reference used and the second is the type of baseline used. These two things become

parameters that can show the integration of determining the boundaries of the sea area. From the results of overlay data, several zones in the provincial sea are outside the territorial sea, which will result in activity permits for local fishermen in Indonesia. The lack of clarity in the permits of the fishermen's operational areas will hinder the freedom of local fishermen to find fish. When the limited fishing area has a clear status, the impact on the fish catch produced is increasingly limited. This is a problem that must be resolved by the Government to improve the welfare of local fishermen.

Possible setbacks in the future may appear due to the discontinuity between the sea boundaries of the Central Indonesian Government and its administrative subdivisions. The stakeholders will have to cooperate to resolve the issues that are currently present, regarding the technical aspect of sea boundary delineation in Indonesian law. These discrepancies lead to multiple misinterpretations and a difference in implementations of the regulations. This condition is also worsened by the overlapping of stakeholders, with each of them having their own rights and authorities. The ideal solution is to review the existing overlapping and redundant regulations.

The new regulations will need to consider the correlation between the authorities and responsibilities of multiple stakeholders, in order to create a common outlook in solving the issue. Indonesia has ratified the UNCLOS III and, therefore, should provide the international standard of procedures that are interpreted by various sectors and activities that are carried out in the sea. The technical aspect of these activities can refer to derivative technical documents of the UNCLOS III that are issued by the International Hydrographic Organization. Each institution must refer to these documents as its primary guidance in order to prevent further discrepancies in its implementation.

This study is aimed at elaborating the consequences of having different methods and principles for determining sea boundaries between the international and national (regional or administrative) levels, which causes discrepancies in the process of governing the area or states. For states that implement some form of decentralization in governing their territories, the differing reference of sea boundary delineation becomes a central issue that is directly related to the integrated development and planning of the area, as well as the coordination that takes place between the stakeholder organizations on-site. The allocation of authority in a decentralized government may create differences in methods and preferences between the central and provincial governments on regulating and developing policies for their respective coastal and sea areas. This study is aimed at giving recommendations regarding points of concern and how to resolve issues that are caused by discrepancies between a sovereign state's central and provincial governments, in determining their respective sea boundaries.

From an international standpoint, the issue regarding the integration of coastal and sea area management is one of urgency, where synchronization and cooperation in the maritime sector between multiple sovereign states are needed. Another point that is worth noting is how the concept of the principles of coastal and sea area management are integrated into the governmental system that takes place in a certain state. The integration process will certainly differ throughout multiple states. This research discusses the Indonesian perspective, in studying examples set by foreign states around the world, on implementing the concept of determining and delineating sea boundaries, both international and regional or administrative, at the intra-state level.

The findings of this study show that as an example, the United States of America has a different method and approach in delineating sea boundaries for its first-level subdivisions. As a federal state, the United States of America implements a uniform standard reference that all states must adhere to in delineating sea boundaries. On the contrary, Indonesia, which is officially a unitary state, implements overlapping and different references in delineating sea boundaries that are used by state institutions at the central level compared to those at the regional/municipal level.

## 5. Conclusions

The integration of sea boundaries between the province and state levels is represented by the integration of physical parameters related to coastal area planning. These parameters include the following:

1.  Coastline. Technical aspects are very influential in the problem of integration of the sea boundaries between provinces and states. The coastline is used as a reference to determine the zoning boundaries that will be set in the coastal area.
2.  Data sources. The right data source can minimize the differences in position on the map presented.
3.  Consistency of statutory references and their derivatives. Based on the latest regulations governing coastal area planning, it is necessary to classify or group rules with the same trend. Agencies in charge of coastal area management need to formulate technical parameters that can be used together.

To resolve the process of implementing Marine Spatial Planning (MSP) and/or Integrated Coastal Zone Management (ICZM), there must be a clarity and a uniform reference between the state and provincial sea boundaries; otherwise, it will create an overlap of authorities between those in the state and province. The challenge ahead is how to harmonize the perceptions in coastal area management from various institutions to achieve the implementation of an established ICZM. The recommendation offered is a review of regulations related to the management of coastal areas, by combining all the main functions of the institution, so that an overlap or gap in authority between institutions can be identified early. The certainty and supremacy of law, regarding the rights and responsibilities of stakeholder organizations in coastal areas without overlapping tasks or duties, will create an optimal environment for the growth and development of the coastal area itself. By maintaining law and order, there will be no coastal areas that are claimed by the jurisdiction of multiple organizations, and no coastal areas that are left behind, underdeveloped due to a lack of interest from stakeholders.

This research is limited to data processing, with various data sources and data resolution. To get optimal results, high-resolution data is needed to improve the spatial accuracy of data analysis. Further research can be done by a broadening of the aspects discussed here, apart from legal, technical, and institutional aspects.

**Author Contributions:** Conceptualization, E.D.; Methodology, A.P.P.; Resources, D.K. and K.Y.; Writing—review & editing, M.M.J. All authors have read and agreed to the published version of the manuscript.

**Funding:** This research was funded by Research, Community Service, and Innovation Program at Bandung Institute of Technology.

**Institutional Review Board Statement:** Not applicable.

**Informed Consent Statement:** Not applicable.

**Data Availability Statement:** Not applicable.

**Acknowledgments:** We thank the Research, Community Service, and Innovation Program at Bandung Institute of Technology for providing this research funding. We appreciate our colleagues Nafandra Syabana Lubis, Ben William Rogers, Candida A.D.S. Nusantara, Aulia Rahma, Ihsan, Mei Handayani, and Gistya Chairunisa, who provided insight and expertise that greatly assisted the research. We appreciate the editor and three anonymous reviewers for their constructive comments, which helped us to improve this article.

**Conflicts of Interest:** The authors declare no conflict of interest.

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
