# Peer review of "The Concept of Integration between State and Provincial Sea Boundaries in Indonesia"

_sustainability, doi:10.3390/su14031659_

Round 1

Reviewer 1 Report

Suggestions and Comments:

- Specify in the title the country or geographical area to which the survey refers,

- Explain more clearly (lines 28-29) what the difference is between a sectoral interest and a sectoral authority with regard to the control of natural resources,

- Text repeat: the paragraph between lines 28 and 38 is identical to the one between lines 39 and 49,

- The introduction needs to explain in more detail the circumstances and the problem with which the inquiry is concerned. It would be appropriate to outline the current practice (legal aspect and spatial planning aspect) of establishing sea boundaries in this country which has led to the current unresolved situation. It is necessary to mention some other examples of how these challenges have been overcome and similar approaches to solving them. In addition to the US case already mentioned, I suggest that you consider some additional studies in this field,

- Before the methodology chapter, it is necessary to define more precisely the research questions that the research addresses,

- In the methodology chapter you state some basic principles from the ICZM Protocol. It is not clear from this text what these are and how you have used them in your research. More descriptive explanations are needed. The methodology chapter needs to be structured more analytically so that the progression of the research is clear through individual stages and steps (e.g. data collection, data analysis, data processing and results etc.). In this chapter, it should be very clear what exactly you did at a particular stage of the research and for what purpose (according to the research questions),

- The results (from line 135 onwards) should be presented systematically and in accordance with the steps of analytical work presented in the methodology,

- The discussion (from line 168 onwards) should answer in detail the questions posed. I suggest that in the above solutions you also refer to the experience of other countries or other practices in developing concepts of integration of sea boundaries.

- General message of the article: The article only talks about existing problems and possible solutions in the country, but perhaps the authors can also identify some existing comparative advantages or previous positive experiences that other countries (in the wider region) do not have. I suggest you look at similar challenges in other examples. This is a very topical issue at the moment, for example in Europe, where the traditional maritime countries of the Mediterranean are in the process of creating a new generation of maritime spatial plans (in line with the MSP Directive). There is much literature on this topic, referring to previous experiences in defining authorities, defining boundaries, coordinating conflicts and interests on the coast and the sea, implementing the ICZM Protocol, etc. (see, e.g., González et al., Ocean Coastal Management, 52, 2009; ÄŒok et al., Water, 13, 2021; etc.).

The article should also highlight more clearly the importance of integrating sea boundaries from the perspective of sustainable spatial planning, which is the red thread of the journal.

Reviewer 2 Report

This draft article suffers from some problems. To begin with, it is only on line 57 that the reader is informed about the geographical focus of the study. Much earlier, reference is made to specific legislation, without any indication of the nationality of the legislation. Reference is also made to the US before even mentioning that the study focuses on (a part of) Indonesia. And the point of comparing Indonesia with the US is never revealed.

The introductory part largely focuses on problems (although never actually specified/exemplified) caused by sectoral legislation, and ‘sectoral contestation of control’ that is ‘due to overlapping sectoral legislation’. However, the actual study that is carried out is seemingly not at all concerned with sectoral (= ‘relating to a sector of the economy’) dimensions, but with geographical discrepancies relating to the mandates of national and regional authorities linked to inconsistent sea boundaries.

The introduction thus never properly introduces or makes the case for the study that is actually carried out.

MSP is defined in a way (line 79) that differs significantly from well-established definitions, such as that developed by UNESCO. This calls for clarification and exemplifies the fact that the study is insufficiently situated in the existing (and large) MSP literature.

In the introductory parts, there is a focus on policy dimensions. However, in the results and discussions sections this is almost completely absent. Beyond a very brief mentioning of ‘potential abuse of local governments' authority in marine areas outside the territorial sea’ and ‘overlapping authority between the Central Government and the Provincial Government’ there is no deliberation or actual discussion about the policy challenges that may accrue from the discrepancies in the special definition of coastal/maritime areas. There is also no discussion about the nature of the different mandates that are affected or the nature of the different zones at issue. The lasting impression is that the article actually has a very technical focus – which is also confirmed by the statement that ‘this study uses a quantitative research methodology by conducting comparative analysis and correlation analysis. This would need to be clearly reflected from the outset so as not to raise expectations of an analysis of actual management conflicts or problems caused by discrepancies between different sea boundaries.

The language is partly in need of significant editing. 

See eg the sentence ‘Lack of understanding of coastal resources nature and function as a life support system for land and sea areas has resulted in a strong sectoral ego in implementing its policies.’ Firstly, it is utterly unclear what ‘its’ refers to. Secondly, the exact same sentence occurs twice in the text (lines 31-32 and 42-43).

Reviewer 3 Report

The Concept of Integration Between State and Province Sea Boundaries The main outcomes of this work are that: 1 Unambiguous definition of the coastline is needed. 2 Sources of positional data need to be defined and integrated 3 The legal field needs to be clear, for example overlapping regional and national authorities, and use of different tidal heights in different legislation. All these are absolutely true. However, none of them are new observations. They may be in Indonesia, but not elsewhere in the world. The limited reference list and lack of international literature may reflect narrowness of view, concentrating on Indonesia. Further, the language used requires very great improvement. Many parts of the text are currently ambiguous or not understandable.

Round 2

Reviewer 1 Report

In the new version, some justifications are now quite extensive, especially in the Materials and Methods section.

It also needs proofreading as some sentences are still repeated.

Reviewer 2 Report

The changes and additions made have improved the text. My main concerns now are the language and some formal issues. There are sentences that I really struggle to understand. 

Some examples:

lines 31-33: 'Activities in the coastal area are managed by cross-sectoral institutions, each of which is interested in carrying out their activities. In its implementation, there is sectoral contestation of control of coastal natural resources and environmental services. This is due to 33 overlapping sectoral legislation.' What does 'its implementation' refer to here?

Lines 42-44: ‘Lack of understanding of coastal resources nature and function as a life support system for land and sea areas has resulted in a strong sectoral ego in implementing its policies.’ What does ‘its’ relate to here?

Text on lines 31-33 is identical to text on lines 55-57.

Line 176, text refers to definition of MSP, but no source is provided.

Lines 450-452: ‘The lack of overlap and the gap in authority will eliminate the impacts felt by the perpetrators of activities in coastal areas with more clarity on the rights and obligations that they need to carry out.’ I just don’t understand this sentence.

There are several acronyms that never seem to be spelled out in the text. 

Reviewer 3 Report

The authors have added text to justify the work. I remain doubtful whther it is of significant international interest. The language requires further careful review.
